# A Secure Authentication Protocol Supporting Efficient Handover for UAV

Kang Wen [ID], Shengbao Wang * [ID], Yixiao Wu, Jie Wang, Lidong Han and Qi Xie [ID]

Key Laboratory of Cryptography of Zhejiang Province, Hangzhou Normal University, Hangzhou 311121, China; kwenlsj@163.com (K.W.); wuyixiao@stu.hznu.edu.cn (Y.W.); jwang@stu.hznu.edu.cn (J.W.); ldhan@hznu.edu.cn (L.H.); qixie@hznu.edu.cn (Q.X.)
* Correspondence: shengbaowang@hznu.edu.cn

**Abstract:** Unmanned Aerial Vehicles (UAVs) are increasingly pivotal in operations such as flood rescue, wildfire surveillance, and covert military endeavors, with their integration into the Internet of Things (IoT) networks broadening the scope of services they provide. Amidst this expansion, security concerns for UAVs have come to the forefront, particularly in open communication environments where they face authentication challenges and risks of sensitive data, including location information, being exposed to unauthorized parties. To address these issues, we propose a secure and lightweight authentication scheme that combines the use of anonymity mechanisms and Physical Unclonable Functions (PUFs). Specifically, we employ pseudo- and temporary identities to maintain the anonymity of UAVs, while also utilizing PUF technology to strengthen the security of Ground Station Servers (GSSs) against physical threats. Rigorous validation through ProVerif and the Random Oracle (ROR) Model indicates our scheme's superior performance over existing protocols in terms of both efficiency and security.

**Keywords:** Unmanned Aerial Vehicle; handover authentication; ProVerif; random oracle model

## 1. Introduction

Unmanned Aerial Vehicle technology has become an indispensable part of the modern scientific and technological landscape, revolutionizing numerous sectors through its versatile applications. These UAVs, commonly known as drones, offer substantial benefits in diverse areas such as military operations, commercial ventures, environmental monitoring, and efficient traffic management [1,2]. The military sector leverages UAVs for surveillance, reconnaissance, and targeted operations, whereas commercial applications range from aerial photography to logistics and delivery services. In environmental monitoring, UAVs play a crucial role in tracking wildlife, assessing disaster zones, and gathering climate data. Similarly, in traffic management, they assist in congestion analysis and accident response.

As UAV technology continues to advance, significant enhancements in performance metrics, including extended flight duration, improved payload capacity, and advanced navigational systems, are observed. These improvements not only broaden the scope of UAV applications but also suggest a future where UAVs could become integral to everyday life and industrial processes. However, alongside these advancements, the rapid evolution of UAV technology introduces notable security challenges [3]. A critical concern involves UAVs' reliance on wireless communication links with GSSs. These links are essential for operational control, data transmission, and firmware updates. The inherent openness of these wireless channels makes UAVs vulnerable to a range of network security threats, including eavesdropping, data interception, and unauthorized access [4]. A particular issue arises from the need for UAVs to frequently switch between GSS domains due to their high mobility and the limited coverage of each GSS. This necessitates the reestablishment of

secure connections, which in turn requires efficient and secure authentication protocols. Traditional authentication methods are often too cumbersome for UAV use, introducing unacceptable delays and overhead in high-speed mobile environments [5]. Therefore, developing lightweight yet robust authentication protocols specifically tailored for UAVs is pressing.

Current UAV authentication protocols face several challenges, exacerbating the security risks [6]. These challenges include vulnerability to impersonation attacks, where malicious entities mimic legitimate UAVs; lack of anonymity, which could compromise the confidentiality of UAV operations; the risk of physical capture, which could lead to unauthorized access to sensitive data; and substantial computational and communication overheads, impractical in UAV contexts where resources are limited. Therefore, addressing these challenges to develop an effective, lightweight, and secure UAV handover authentication scheme is not only complex but also critical for the safe and efficient operation of UAVs in various domains. The development of such a scheme would require a multi-faceted approach, considering the unique characteristics of UAV operations, the dynamic nature of their environments, and the balance between security and performance.

The main contributions of this paper can be summarized as follows:

1.  We have proposed a two-part handover authentication protocol tailored for UAV scenarios, distinguished by its lightweight and secure framework. This protocol is divided into initial authentication and subsequent handover authentication phases, enabling UAVs to achieve rapid and efficient verification with GSS following a successful initial authentication. The design significantly reduces overhead during the handover process, addressing a critical need in UAV operations for swift and secure authentication mechanisms.

2.  Our protocol demonstrates exceptional defense capabilities against a wide array of common cyber threats, further augmented by its provision for user anonymity and resilience against physical attacks. The security of our protocol was rigorously validated using advanced evaluation tools like ProVerif and the ROR model, ensuring its robustness and reliability for UAV applications. This outcome directly stems from the protocol's design principles and operational mechanics, illustrating its comprehensive security advantages over existing solutions.

3.  Moreover, our protocol's design and implementation have been shown to outperform existing authentication protocols in terms of reducing both communication and computational overheads. This performance efficiency not only underscores the protocol's suitability for UAV applications but also positions it as a more effective alternative to current authentication methods. The analysis and comparative assessment highlight how the protocol's innovative features contribute directly to operational efficiency, making it an advantageous choice for UAV scenarios.

### 1.1. Related Work

The inception of Unmanned Aerial Vehicles (UAVs) was pioneered by Gharibi et al. [7], introducing a hierarchical network control architecture for these systems. UAVs are instrumental in delivering a plethora of services, such as package delivery, traffic surveillance, and disaster response, significantly boosting work efficiency, enhancing life quality, and fostering new commercial ventures. However, the reliance of UAV communications on public channels introduces substantial security vulnerabilities, endangering data integrity and privacy. Such breaches pose dire consequences, prompting the development of robust authentication frameworks tailored for UAV ecosystems.

Deebak et al. [8] proposed a lightweight, privacy-centric scheme aimed at reducing computational burdens through autonomous knowledge acquisition. Despite its innovations, the scheme's resilience against Global Satellite System (GSS) impersonation attacks remains insufficient [9]. Cho et al. [10] devised a bespoke authentication mechanism for UAVs, facilitating session key generation and verification by GSSs for drones on predetermined routes. However, vulnerabilities to privileged insider and verification table

leakage attacks were exposed by Jan et al. [11], who then recommended a symmetric key authentication strategy to mitigate these vulnerabilities.

Further exploration by Zhang et al. [12] yielded an authentication protocol leveraging hash and XOR operations, though Chaudhary et al. [13] later identified susceptibilities to several forms of attacks, including privileged insider and smart card theft. Hussain et al. [14] introduced an elliptic curve-based authentication scheme, enhancing user-UAV communication security within designated zones. Yet, it was found vulnerable to drone impersonation and session key compromises by Zhang et al. [15]. These methodologies, however, overlooked the critical handover process necessary for extended UAV flights.

Addressing long-distance communication challenges, Kumar et al. [16] in 2018 advanced a handover protocol integrating device and base station consistency, albeit without considering computational limitations due to bilinear pairings dependency. Son et al. [17] in 2022 innovated a blockchain-based protocol facilitating UAV-GSS authentication post-initial verification, albeit susceptible to various attack vectors [18]. Babu et al. [19] developed a PUF-based protocol for seamless UAV charging, yet it remains exposed to replay attacks and lacks forward secrecy. Kwon et al. [20] introduced a handover scheme vulnerable to physical assaults and burdened by excessive overhead from GSS involvement in the process. In response, Khalid et al. [21] unveiled an efficient, anonymous handover authentication protocol in 2023, utilizing AES-RSA for heightened security, albeit with concerns over computational demands. Ren et al. [22] subsequently proposed a comprehensive, novel handover protocol for UAV applications, incorporating three distinct authentication phases. While innovative, the protocol's complexity and communication demands may impede practical implementation in UAV operations.

### *1.2. Organization*

In Section 2, we introduce the preliminaries of the protocol. Section 3 presents the details of our proposed protocol. In Section 4, we perform an informal security analysis of the proposed protocol. Section 5 uses the Random Oracle Model and the ProVerif formal verification tools to verify the security of the protocol. Additionally, Section 6 provides a comprehensive analysis of the protocol's performance. Finally, we draw our conclusions in Section 7.

## 2. Preliminaries

In this section, we present an overview of the preliminaries, encompassing elliptic curve cryptography, physical unclonable functions, the system model, and the threat model.

### *2.1. Elliptic Curve Cryptography*

Consider $F_p$ as a finite field where $P$ is a prime number. Within $F_p$, define $E$ ($a$, $b$): $y^2 = x^3 + ax + b$, where $a, b \in F_p$ and $4a^3 + 27b^2 \bmod q \neq 0$. Let G be a cyclic group of prime order $q$, with $P$ as the generator point.

**Definition 1.** *Elliptic Curve Discrete Logarithm Problem (ECDLP): For given points $P$, $Q \in G$, where $Q = s \cdot P$, it is computationally difficult to determine $s$ from $Q$ within polynomial time.*

**Definition 2.** *Elliptic Curve Computational Diffie-Hellman Problem (ECCDH): Given points $P$, $a \cdot P$, $b \cdot P \in G$, it is challenging to compute $a \cdot b \cdot P$ within polynomial time.*

### *2.2. Physical Unclonable Function*

A PUF is a random function derived from the physical properties of a device. It exploits minor manufacturing variations in chips to generate unique keys. A PUF can be expressed as $R = PUF(C)$, where $C$ represents the challenge value and $R$ the response value. PUFs are characterized by two main properties:

- Consistency: The PUF consistently produces the same output for a given input.

- Uniqueness: Each semiconductor device has a unique PUF response or output. This uniqueness is derived from the specific manufacturing variations, making it extremely difficult for two devices to have identical PUF outputs.

### 2.3. System Model

Illustrated in Figure 1, we present an overview of the system model for our proposed protocol, which comprises three entities:

1. UAV: The UAV is with limited computing and storage resources. It communicates with GSSs to receive control commands and transmit sensor data. During handovers, it authenticates with the new GSS.
2. GSS: GSS provide communication links and control interfaces for UAVs. As UAVs move between GSS coverage areas, they may switch between GSS, necessitating authentication with the new GSS. GSS possess greater computational power but could be vulnerable to external attacks.
3. RA: The Registration Authority (RA) serves as a trusted third party. It issues cryptographic credentials such as certificates to UAVs during registration. The RA also shares essential public parameters with UAVs and GSS to facilitate the authentication process.

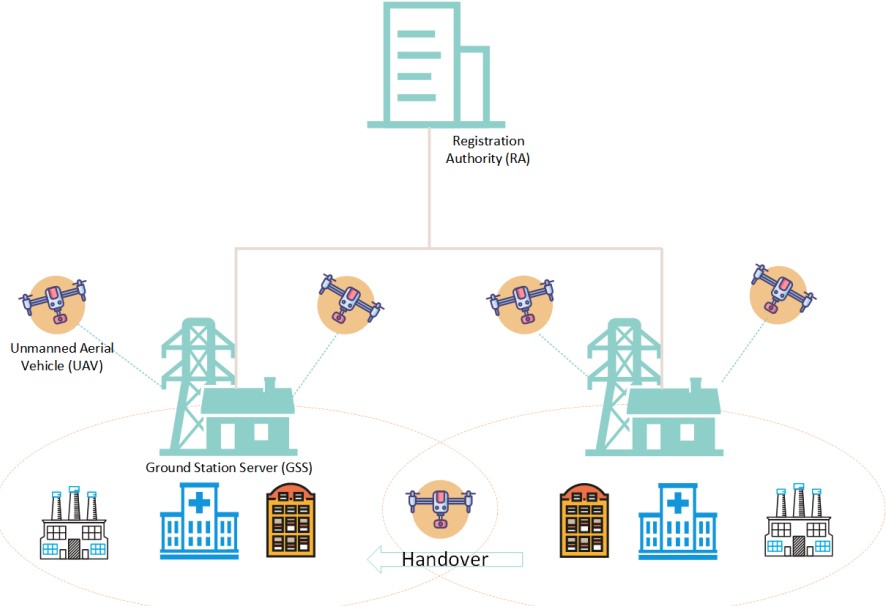

**Figure 1.** Systematic architecture of proposed scheme.

### 2.4. Adversary Model

The Dolev-Yao threat model [23], introduced by Dolev and Yao in 1983, is a cornerstone in cybersecurity. This seminal model distinctively delineates the security protocol from the specific cryptographic algorithms it utilizes. Its primary application is the analysis of a protocol's security under the assumption of an ideal cryptographic system. This framework allows for the evaluation and validation of our proposed authentication and key agreement protocol for UAV communication, irrespective of the subsequent symmetric key encryption and decryption processes.

The Dolev-Yao model rigorously defines the adversary, represented as $\mathcal{A}$, and their potential attack methods, which include:

1. $\mathcal{A}$'s ability to eavesdrop, intercept, delete, or alter messages over insecure wireless channels. However, they cannot modify messages sent through secure channels.
2. $\mathcal{A}$'s capability to store intercepted messages and replay them to legitimate entities such as UAVs and ground station server, and to fabricate and send false messages to impersonate legitimate parties [24].

3.  $\mathcal{A}$'s potential to seize network nodes, like GSS, and extract cryptographic keys or other information through physical attack [25].

This model specifies $\mathcal{A}$'s capabilities in the context of UAV communication networks, utilizing the Dolev-Yao assumptions to assess the security and robustness of our novel authentication protocol.

## 3. The Proposed Handover Authentication Scheme

In this section, we will describe our proposed lightweight and secure handover authentication scheme for UAV. The specific scheme has four phases: system initialization phase, registration phase, UAV initial authentication phase and UAV handover authentication phase.

1.  System initialization phase: This phase completes the generation and publication of public parameters, including the RA's public key, private key, and appropriate hash functions.
2.  Registration phase: The registration phase includes the registration of the GSS and the registration of the UAV. The GSS protects the registered information through the PUF, and the UAV hides the registration information through biometrics.
3.  UAV initial authentication phase: This phase mainly completes the UAV's initial authentication in the GSS.
4.  Handover authentication phase: This phase mainly completes authentication and key negotiation when the UAV is moving from one GSS to another GSS network.

### 3.1. System Initialization Phase

The RA builds an elliptic curve $E_p(a, b)$ using $P$ as the generator within the group $\mathbb{G}$. In addition to selecting its own secret key $SK_{RA}$ and corresponding public key $PK_{RA}$, the RA uses the one-way hash function $h(\cdot)$.

### 3.2. Registration Phase

The registration phase encompasses enrollment of the GSSs and UAVs. It has two parts.

#### 3.2.1. GSS Registration Phase

In the GSS Registration Phase, as illustrated in Table 1, each $GSS_i$ selects a unique identity $GID_i$, and securely transmits it to the RA over a secure channel. RA verifies the identity's uniqueness. Upon confirmation of uniqueness, RA generates a random number $b_i$, and computes $GSS_i$'s private key $SK_{GSS_i} = h(GID_i||SK_{RA}||b_i)$, and the corresponding public key, $PK_{GSS_i}$ calculated as $SK_{GSS_i} \cdot P$. Simultaneously, a shared secret $k = h(RID||SK_{RA})$, is established between RA and $GSS_i$. RA confidentially transmits the values $k, SK_{GSS_i}$ to $GSS_i$ and publicly discloses $PK_{GSS_i}$ while securely storing the tuple $(GID_i, b_i)$ in its memory.

Upon receiving $\{k, SK_{GSS_i}\}$, $GSS_i$ initiates a challenge-response mechanism, generating a challenge $Cha_i$ and computing the corresponding response, $Res_i = PUF(Cha_i)$. Subsequently, $GSS_i$ computes $Y_i = SK_{GSS_i} \oplus h(GID_i||Res_i)$ and $V_i = k \oplus h(GID_i||SK_{GSS_i})$, while retaining the values $\{Y_i, V_i, Cha_i\}$.

#### 3.2.2. UAV Registration Phase

In the UAV Registration Phase, as illustrated in Table 2, each $UAV_i$ selects a unique identity $UID_i$ and securely transmits it to the RA. RA verifies the uniqueness of the identity. Upon confirmation of uniqueness, RA generates a random number $a_i$, calculates $A_i = a_i \cdot P$, and derives $UAV_i$'s pseudo-identity $PID_i = h(UID_i||A_i)$. Additionally, it computes $d_i = a_i + PID_i \cdot SK_{RA}$. RA securely transmits the values $\{A_i, d_i\}$ to $UAV_i$ via a secure channel while securely storing the tuple $(UID_i, a_i)$ in its memory.

Upon reception, $UAV_i$ recalculates $PID_i$ as $h(UID_i||A_i)$ and verifies the equation $d_i \cdot P \overset{?}{=} A_i + PID_i \cdot PK_{RA}$. Successful verification prompts $UAV_i$ to input its biometric information $Bio_i$. It computes $Gen(Bio_i) = (\sigma_i, \tau_i)$, $F_i = A_i \oplus h(\sigma_i)$, $G_i = d_i \oplus h(A_i||PID_i)$, and $H_i = h(A_i||d_i||PID_i)$. Finally, $UAV_i$ stores the values $\{F_i, G_i, H_i, Rep(\cdot)\}$.

**Table 1.** GSS registration phase.

| $GSS_i$ | RA |
|---|---|
| Chooses $GID_i$ | |
| $\xrightarrow{\{GID_i\}}$ Secure channel | |
| | Verify the uniqueness of $GID_i$ |
| | Select $b_i$ |
| | Calculates $SK_{GSS_i} = h(GID_i \| SK_{RA} \| b_i)$ |
| | $PK_{GSS_i} = SK_{GSS_i} \cdot P$ |
| | $k = h(RID \| SK_{RA})$ |
| | Stores $(GID_i, b_i)$ in secure memory |
| $\xleftarrow{\{k, SK_{GSS_i}\}}$ Secure channel | |
| Generates a challenge $Cha_i$ | |
| $Res_i = PUF(Cha_i)$ | |
| $Y_i = SK_{GSS_i} \oplus h(GID_i \| Res_i)$ | |
| $V_i = k \oplus h(GID_i \| SK_{GSS_i})$ | |
| Stores $\{Y_i, V_i, Cha_i\}$ | |

**Table 2.** UAV registration phase.

| $UAV_i$ | RA |
|---|---|
| Chooses $UID_i$ | |
| $\xrightarrow{\{UID_i\}}$ Secure channel | |
| | Verify the uniqueness of $UID_i$ |
| | Selects $a_i$ |
| | Calculates $A_i = a_i \cdot P$ |
| | $PID_i = h(UID_i \| A_i)$ |
| | $d_i = a_i + PID_i \cdot SK_{RA}$ |
| | Stores $(UID_i, a_i)$ in secure memory |
| $\xleftarrow{\{A_i, d_i\}}$ Secure channel | |
| Computes $PID_i = h(UID_i \| A_i)$ | |
| Verify $d_i \cdot P \overset{?}{=} A_i + PID_i \cdot PK_{RA}$ | |
| Input $Bio_i$ | |
| $Gen(Bio_i) = (\sigma_i, \tau_i)$ | |
| $F_i = A_i \oplus h(\sigma_i)$ | |
| $G_i = d_i \oplus h(A_i \| PID_i)$ | |
| $H_i = h(A_i \| d_i \| PID_i)$ | |
| Stores $(F_i, G_i, H_i, Rep(\cdot))$ | |

*3.3. UAV Initial Authentication Phase*

When $UAV_i$ initially enters the coverage area of GSS, it is required to undergo an initial authentication process, as illustrated in Table 3. The detailed process unfolds as follows.

1. The $UAV_i$ inputs the user's biological information $Bio_i^*$, recovers $\sigma_i^* = Rep(Bio_i^*, \tau_i)$, then calculates $A_i^* = h(\sigma_i^*) \oplus F_i$, $PID_i^* = h(UID_i \| A_i^*)$, computes $d_i^* = G_i \oplus h(A_i^* \| PID_i^*)$, and verifies $H_i$ to check if it equals $h(A_i^* \| d_i^* \| PID_i^*)$.

2. If the verification is successful, $UAV_i$ selects a random number $c_i$ and the current timestamp $T_1$, and then calculates $P_{it} = c_i \cdot PK_{GSS_i}$, $P_i = c_i \cdot P$, $W_1 = PID_i \oplus h(GID_i \| P_{it})$, $W_2 = d_i \oplus h(PID_i \| T_1 \| P_{it})$, and $W_3 = h(PID_i \| d_i \| A_i \| P_{it} \| T_1)$. It then sends the computed tuple $(W_1, W_2, W_3, P_i, T_1)$ to $GSS_i$.

3. Upon receiving the values, $GSS_i$ verifies the freshness of $T_1$, calculates $Res_i = PUF(Cha_i)$, $SK_{GSS_i} = Y_i \oplus h(GID_i \| Res_i)$, $P_{it} = SK_{GSS_i} \cdot P_i$, $PID_i = W_1 \oplus h(GID_i \| P_{it})$,

$d_i = W_2 \oplus h(PID_i \| T_1 \| P_{it})$, and $A^* = d_i \cdot P - PID \cdot PK_{RA}$. It then verifies that $W_3$ is equal to $h(PID_i \| d_i \| A^* \| P_{it} \| T_1)$.

4. If the validation is successful, $GSS_i$ selects random numbers $e_i$, $n_i$, and a timestamp $T_2$, and calculates $K_{it} = e_i \cdot P_i$, $K_i = e_i \cdot P$, $TID_i = PID_i \oplus h(n_i)$, $W_4 = TID_i \oplus h(K_{it} \| PID_i)$, $SK_{it} = h(K_{it} \| PID_i \| TID_i)$, $k = V_i \oplus h(GID_i \| SK_{GSS_i})$, $Q_i = h(TID_i \| k)$, $W_5 = Q_i \oplus h(K_{it})$, and $W_6 = h(SK_{it} \| Q_i \| T_2)$. $GSS_i$ then sends the values $(W_4, W_5, W_6, K_i, T_2)$ to $UAV_i$.

5. Based on the received values, $UAV_i$ checks the freshness of $T_2$ and calculates $K_{it} = c_i \cdot K_i$, $TID_i = W_4 \oplus h(K_{it} \| PID_i)$, $Q_i = W_5 \oplus h(K_{it})$, and $SK_{it} = h(K_{it} \| PID_i \| TID_i)$. Finally, $UAV_i$ verifies $W_6$ to confirm if it equals $h(SK_{it} \| Q_i \| T_2)$. Consequently, $UAV_i$ completes authentication with $GSS_i$, securing the session key $SK_{it}$ and a temporary identity $TID_i$.

**Table 3.** UAV initial authentication phase.

| $UAV_i$ | $GSS_i$ |
|---|---|
| Input $Bio_i^*$ | |
| $\sigma_i^* = Rep(Bio_i^*, \tau_i)$ | |
| $A_i^* = h(\sigma_i^*) \oplus F_i$ | |
| $PID_i^* = h(UID_i \| A_i^*)$ | |
| Computes $d_i^* = G_i \oplus h(A_i^* \| PID_i^*)$ | |
| Checks $H_i \overset{?}{=} h(A_i^* \| d_i^* \| PID_i^*)$ | |
| Generates $c_i$ and $T_1$ | |
| $P_{it} = c_i \cdot PK_{GSS_i}$ | |
| $P_i = c_i \cdot P$ | |
| $W_1 = PID_i \oplus h(GID_i \| P_{it})$ | |
| $W_2 = d_i \oplus h(PID_i \| T_1 \| P_{it})$ | |
| $W_3 = h(PID_i \| d_i \| A_i \| P_{it} \| T_1)$ | |
| $\xrightarrow{(W_1, W_2, W_3, P_i, T_1)}$ | |
| | Checks $T_1$ |
| | $Res_i = PUF(Cha_i)$ |
| | $SK_{GSS_i} = Y_i \oplus h(GID_i \| Res_i)$ |
| | Computes $P_{it} = SK_{GSS_i} \cdot P_i$ |
| | $PID_i = W_1 \oplus h(GID_i \| P_{it})$ |
| | $d_i = W_2 \oplus h(PID_i \| T_1 \| P_{it})$ |
| | Calculates $A^* = d_i \cdot P - PID \cdot PK_{RA}$ |
| | Checks $W_3 \overset{?}{=} h(PID_i \| d_i \| A^* \| P_{it} \| T_1)$ |
| | Generates $e_i, n_i$ and $T_2$ |
| | $K_{it} = e_i \cdot P_i$ |
| | $K_i = e_i \cdot P$ |
| | $TID_i = PID_i \oplus h(n_i)$ |
| | $W_4 = TID_i \oplus h(K_{it} \| PID_i)$ |
| | $SK_{it} = h(K_{it} \| PID_i \| TID_i)$ |
| | $k = V_i \oplus h(GID_i \| SK_{GSS_i})$ |
| | $Q_i = h(TID_i \| k)$ |
| | $W_5 = Q_i \oplus h(K_{it})$ |
| | $W_6 = h(SK_{it} \| Q_i \| T_2)$ |
| $\xleftarrow{(W_4, W_5, W_6, K_i, T_2)}$ | |
| Checks $T_2$ | |
| Computes $K_{it} = c_i \cdot K_i$ | |
| $TID_i = W_4 \oplus h(K_{it} \| PID_i)$ | |
| $Q_i = W_5 \oplus h(K_{it})$ | |
| $SK_{it} = h(K_{it} \| PID_i \| TID_i)$ | |
| Checks $W_6 \overset{?}{=} h(SK_{it} \| Q_i \| T_2)$ | |

*3.4. UAV Handover Authentication Phase*

After the successful authentication of the $UAV_i$ and $GSS_i$, when the $UAV_i$ enters the coverage of $GSS_j$, the $UAV_i$ and $GSS_j$ need to complete a new authentication. This handover authentication process is described in Table 4, and the process is shown below.

1. The $UAV_i$ generates a random number $m_i$ and a timestamp $T_3$, then calculates $W_7 = h(PID_i||TID_i||GID_j||T_3) \oplus m_i$, $W_8 = Q_i \oplus PID_i$, $W_9 = h(m_i||PID_i||TID_i||T_3)$. Subsequently, the $UAV_i$ sends $(TID_i, W_7, W_8, W_9, T_3)$ to $GSS_j$.

2. Once the above information is received, $GSS_j$ first checks the freshness of the $T_3$, and if the test passes, calculates $Res_j = PUF(Cha_j)$, $SK_{GSS_j} = Y_j \oplus h(GID_j||Res_j)$, $k = V_j \oplus h(GID_j||SK_{GSS_j})$, Computes $PID_i^* = W_8 \oplus h(TID_i||k)$, $m_i^* = W_7 \oplus h(PID_i^*||TID_i||GID_j||T_3)$, and verifies $W_9 \stackrel{?}{=} h(m_i||PID_i^*||TID_i||T_3)$.

3. If the above verification is passed, $GSS_j$ generates random numbers $e_j, n_j$ and a timestamp $T_4$, calculates $TID_j = PID_i \oplus h(n_j)$, $W_{10} = TID_j \oplus h(m_i||PID_i)$, $W_{11} = e_j \oplus h(m_i||TID_j||PID_i)$, $SK_{ij} = h(m_i||e_j||TID_j||PID_i)$, $W_{12} = h(SK_{ij}||T_4)$, and then sends the calculated results $(W_{10}, W_{11}, W_{12}, T_4)$ to $UAV_i$.

4. After receiving the information transmitted by $GSS_j$, the $UAV_i$ verifies the freshness of the $T_4$, then calculates $TID_j = W_{10} \oplus h(m_i||PID_i)$, $e_j = W_{11} \oplus h(m_i||TID_j||PID_i)$, $SK_{ij} = h(m_i||e_j||TID_j||PID_i)$, verifies $M_{13} \stackrel{?}{=} h(SK_{ij}||T_4)$. Through the above calculation, the session key $SK_{ij}$ can be obtained, and a new temporary identity $TID_j$ can be obtained. At this point, the $UAV_i$ completes the handover authentication process.

**Table 4.** UAV handover authentication phase.

| $UAV_i$ | $GSS_j$ |
|---|---|
| Generates $m_i, T_3$ | |
| Calculates | |
| $W_7 = h(PID_i||TID_i||GID_j||T_3) \oplus m_i$ | |
| $W_8 = Q_i \oplus PID_i$ | |
| $W_9 = h(m_i||PID_i||TID_i||T_3)$ | |
| $\xrightarrow{(TID_i, W_7, W_8, W_9, T_3)}$ | |
| | Checks $T_3$ |
| | $Res_j = PUF(Cha_j)$ |
| | $SK_{GSS_j} = Y_j \oplus h(GID_j||Res_j)$ |
| | $k = V_j \oplus h(GID_j||SK_{GSS_j})$ |
| | Computes $PID_i^* = W_8 \oplus h(TID_i||k)$ |
| | $m_i^* = W_7 \oplus h(PID_i^*||TID_i||GID_j||T_3)$ |
| | Checks $W_9 \stackrel{?}{=} h(m_i||PID_i^*||TID_i||T_3)$ |
| | Generates $e_j, n_j$ and $T_4$ |
| | Computes $TID_j = PID_i \oplus h(n_j)$ |
| | $W_{10} = TID_j \oplus h(m_i||PID_i)$ |
| | $W_{11} = e_j \oplus h(m_i||TID_j||PID_i)$ |
| | $SK_{ij} = h(m_i||e_j||TID_j||PID_i)$ |
| | $W_{12} = h(SK_{ij}||T_4)$ |
| $\xleftarrow{(W_{10}, W_{11}, W_{12}, T_4)}$ | |
| Checks $T_4$ | |
| Computes | |
| $TID_j = W_{10} \oplus h(m_i||PID_i)$ | |
| $e_j = W_{11} \oplus h(m_i||TID_j||PID_i)$ | |
| $SK_{ij} = h(m_i||e_j||TID_j||PID_i)$ | |
| Checks $W_{12} \stackrel{?}{=} h(SK_{ij}||T_4)$ | |

## 4. Informal Security Analysis

In this section, we show that our proposed scheme is secure through analysis on various desirable security properties.

### 4.1. Mutual Authentication

In our protocol, $GSS_i$ authenticates $UAV_i$ by verifying the correctness of $A_i \overset{?}{=} d_i \cdot P - PID \cdot PK_{RA}$. Conversely, $UAV_i$ authenticates $GSS_j$ by validating the correctness of $K_{it} = c_i \cdot K_i$. This process ensures mutual authentication between $GSS_i$ and $UAV_i$.

### 4.2. Impersonation Attack

Consider a scenario where an adversary $\mathcal{A}$ attempts to impersonate a UAV. $\mathcal{A}$ intercepts the messages $(W_1, W_2, W_3, P_i, T_1)$ on public channels, where each $W$ is defined by specific cryptographic operations. Despite interception, $\mathcal{A}$ cannot compute $W_1$, $W_2$, and $W_3$ due to the lack of access to $PID_i$. Therefore, our protocol is resilient against UAV impersonation attacks.

### 4.3. Replay Attack

Assume an adversary $\mathcal{A}$ captures previously transmitted messages over public channels. $\mathcal{A}$ may try retransmitting these messages. However, due to the incorporation of a timestamp mechanism in our protocol, which guarantees message freshness, $\mathcal{A}$ cannot generate a session key with the GSS. Consequently, our protocol effectively thwarts replay attacks.

### 4.4. GSS Physical Capture Attack

In our protocol, each GSS is equipped with a PUF and stores $Y_i$, $V_i$, and $Cha_i$, defined by specific cryptographic operations. In the event of a GSS capture by an adversary $\mathcal{A}$, they cannot access the secret parameters $SK_{GSS_i}$ and $k$. Hence, our protocol is safeguarded against GSS physical capture attacks.

### 4.5. MITM Attack

In a man-in-the-middle (MITM) attack scenario, adversary $\mathcal{A}$ intercepts specific messages. However, $\mathcal{A}$ is unable to access crucial secret values and random numbers necessary for generating authentication requests/responses and the session key. This incapacity of $\mathcal{A}$ to derive these critical elements ensures our scheme's resistance to MITM attacks.

### 4.6. Anonymity and Untraceability

During preliminary authentication, a pseudonym $PID_i$ is used, safeguarding the UAV's real identity $GID_i$. This approach ensures the UAV's identity remains anonymous. Furthermore, the usage of a dynamic temporary identity $TID_i$, updated during handover authentication, prevents adversary $\mathcal{A}$ from tracking the UAV, thus ensuring unlinkability.

### 4.7. Perfect Forward Secrecy

Assuming the leakage of long-term private keys of entities, our protocol maintains security. The session key $SK_{it}$, derived through complex cryptographic operations, remains secure due to the ECDLP problem, preventing adversary $\mathcal{A}$ from deducing the random numbers from each session. This design ensures the provision of perfect forward secrecy in our scheme.

## 5. Formal Security Analysis

In this section, we present a formal security proof using a ROR model and utilize the ProVerif formal verification tool to validate the proposed security protocol.

*5.1. Formal Security Analysis under ROR Model*

**Definition 3.** *(Participants): Three parties involved in our protocol: one Unmanned Aerial Vehicle (UAV), one ground station server (GSS) and one registration authority (RA). Each party can have multiple instances, and the i-th instance of UAV and GSS are denoted as $U^i$ and $G^i$, respectively. The verification can output three possible results. The accept state indicates that the input message is valid. The reject state means the input data is incorrect. The $\perp$ state represents that there is no response to the input. The adversary is able to simulate queries to interact with the UVA or GSS. The details of the queries are presented in Table 5.*

**Table 5.** Queries in ROR model.

| Queries | Description |
|---|---|
| Execute($U^i$, $G^i$) | $\mathcal{A}$ can obtain all publicly transmitted information between $U^i$ and $G^i$. |
| Send($U^i$, $G^i$, $m$) | This query simulates an active attack. $\mathcal{A}$ can send messages to $U^i$ and $G^i$, and obtain respective responses. |
| Reveal($U^i$, $G^i$) | $\mathcal{A}$ can get the session keys between $U^i$ and $G^i$. |
| Corrupt($U^i$, $G^i$) | $\mathcal{A}$ can obtain the stored information $\{F_i, G_i, H_i, Rep(\cdot)\}$ and $\{Y_i, V_i, Cha_i\}$ of $U^i$ and $G^i$. |
| Test($U^i$, $G^i$, $r$) | $\mathcal{A}$ selects a session to launch a reveal query. This will generate a random number $r$. When $r = 1$, the actual session key can be obtained by $\mathcal{A}$; when $r = 0$, $\mathcal{A}$ will get a random number with the same length as the session key. |

**Definition 4.** *(Semantic security): $\mathcal{A}$ is permitted to make a single query to the Test($U^i$, $G^i$, $r$) and and multiple other queries to determine the correctness of the return value of Test($U^i$, $G^i$, $r$). $\mathcal{A}$'s advantage in guessing $r$ is defined as $Adv_P = |2Pr[suc(A)] - 1| < \eta$ represents the protocol is secure, where $\eta$ is sufficiently small.*

**Theorem 1.** *The advantage of obtaining the session key in polynomial time by $\mathcal{A}$ is $Adv_P \leq \frac{q_h^2}{2^{l_h}} + \frac{q_s}{2^{l_{bio}-1}} + 2Adv_{PUF} + 2Adv_{ECDLP}$. Where $q_s$, $q_h$, and $q_e$ represent performing the queries Send, Hash and Execute within time t. The hash, transcripts, and biological key have lengths of $l_h$, $l_{bio}$ and n respectively. The advantages of $\mathcal{A}$ in breaking the PUF and ECDLP are $Adv_{PUF}$ and $Adv_{ECDLP}$ respectively.*

**Proof.** The games are defined to simulate the attacks launched by $\mathcal{A}$, and divided from $G_0$ to $G_4$. $\mathcal{A}$ correctly guessing the random number $r$ represents $Win_i (0 \leq i \leq 4)$.

$G_0$: This game simulates the real attack initially launched by $\mathcal{A}$. According to the definition, we obtain:

$$Adv_{G_0} = |2Pr[Win_0 - 1]| \tag{1}$$

$G_1$: This game simulates the *Execute* query to obtain all publicly transmitted messages. Then, $\mathcal{A}$ verifies the session key through the *Reveals* and *Test* queries. Due to the ECDLP, the attacker cannot determine the association between the captured messages and the session key. Hence,

$$Pr[Win_1] = Pr[Win_0] \tag{2}$$

$G_2$: This game simulates hash and transcript collisions. By the Birthday Paradox, the probability of hash collisions is at most $\frac{q_h^2}{2^{l_h}}$. Therefore, we obtain:

$$Pr[Win_2] - Pr[Win_1] \leq \frac{q_h^2}{2^{l_h+1}} + \frac{(q_s + q_e)^2}{2n} \tag{3}$$

$G_3$: This game simulates the *Corrupt* query to obtain stored information $\{F_i, G_i, H_i, Rep(\cdot)\}$ in UAV and $\{Y_i, V_i, Cha_i\}$ in GSS, where $F_i = A_i \oplus h(\sigma_i)$, $\sigma_i$ is the biometric key, $SK_{GSS_i} =$

$h(GID_i||PUF(Cha_i)) \oplus Y_i$. If $\mathcal{A}$ is able to guess the value of $\sigma$ or break the PUF, then $\mathcal{A}$ will be able to access valuable parameters. As a result, we get:

$$Pr[Win_3] - Pr[Win_2] \leq \frac{q_s}{2^{l_{bio}}} + Adv_{PUF} \tag{4}$$

$G_4$: $\mathcal{A}$ can obtain $P_i = c_i \cdot P$ and $K_i = e_i \cdot P$ publicly, which are then used for session key agreement. This game simulates $\mathcal{A}$ solving the ECDH problem. We have:

$$Pr[Win_4] - Pr[Win_3] \leq Adv_{ECDLP} \tag{5}$$

The session key is independently randomly generated, meaning that $\mathcal{A}$ guessing $r$ has the same difficulty as guessing the session key directly. As a result, we have:

$$Pr[Win_4] = \frac{1}{2} \tag{6}$$

Combining the above formulas, we have:

$$\frac{1}{2} Adv_P = |Pr[Win_0] - \frac{1}{2}| \tag{7}$$

$$\leq \frac{q_h^2}{2^{l_h+1}} + \frac{q_s}{2^{l_{bio}}} + Adv_{PUF} + Adv_{ECDLP} \tag{8}$$

$$Adv_P \leq \frac{q_h^2}{2^{l_h}} + \frac{q_s}{2^{l_{bio}-1}} + 2Adv_{PUF} + 2Adv_{ECDLP} \tag{9}$$

$\square$

### 5.2. Formal Verification Using ProVerif

ProVerif is recognized as an automated verification tool adept at handling a variety of cryptographic algorithms, including symmetric and asymmetric encryption, hash functions, digital signatures, and more. It is particularly effective in assessing security properties such as confidentiality, authentication, and other essential attributes. In this section, we utilize ProVerif to evaluate the security of our proposed scheme.

The initial segment presents declarations relevant to the scheme, covering aspects such as message transmission channels, constants, variables, functions, and events. Figure 2 details the definition of a public channel, named *ch1*, utilized for UAV-GSS node communication. This includes the establishment of constants, variables, the hash function $h(\cdot)$, various connection functions, XOR, and ECC operations. The model for the attacker's queries and the events are primarily detailed in Figure 3.

The second segment comprehensively examines the participation process of the UAV, as depicted in Figure 4. Initially, the UAV retrieves stored data following biological verification, sends authentication messages to the GSS, and generates session keys based on the information received from the GSS. The involvement of the GSS is illustrated in Figure 5. This includes decrypting messages using the GSS's private key, verifying the UAV's identity via the RA's public key, generating a temporary identity and session key, and securely transmitting these to the UAV.

Figure 6 displays the outcomes of ProVerif's analysis of our scheme. The results confirm that adversaries are unable to access key parameters necessary for session key computation. As a result, our proposed scheme is validated as secure.

free chn:channel.

type UAV.

type GSS.

type RA.

type key.

type nonce.

type bioinformation.

type timestamp.

free UIDi:bitstring [private]. (*--The identify of UAV.--*)

free bioi:bioinformation [private]. (*--The bioinformation of UAV.--*

free sk_ra:bitstring [private]. (*--The secret parameter of RA.--*)

free sk_gss:bitstring [private]. (*--The secret parameter of GSS.--*)

free k:bitstring [private]. (*--The shared secret of GSSs.--*)

free SKu:bitstring [private].(*--The session key of UAV.--*)

free SKg:bitstring [private].(*--The session key of GSS.--*)

free cj:bitstring [private]. (*-- The secret parameter of GSS.--*)

free di:bitstring [private]. (*-- The secret parameter of UAV--*)

free GIDj:bitstring. (*--The identify of GSS.--*)

free RID:bitstring. (*--The identify of RA.--*)

free P:bitstring. (*--The basic point--*)

free chaj:bitstring. (*--The Challenge value--*)

free uav:UAV.

free ra:RA.

free gss:GSS.

fun Hash(bitstring):bitstring.

fun Gen(bioinformation):bitstring.

fun Rep(bioinformation,bitstring):bitstring.

fun PUF(bitstring):bitstring.

fun bit_timestamp(timestamp):bitstring.

fun key_bit(bitstring):key.

fun bit_nonce(nonce):bitstring.

fun Enc(bitstring,bitstring):bitstring.

fun Dec(bitstring,bitstring):bitstring.

fun EccMul(bitstring, bitstring): bitstring.

fun EccSub(bitstring, bitstring): bitstring.

fun add(bitstring, bitstring) : bitstring.

fun mul(bitstring, bitstring) : bitstring.

fun XOR(bitstring,bitstring):bitstring.

equation forall x:bitstring,y:bitstring;

XOR(XOR(x,y),y)=x.

fun pufuctionuction(bitstring) : bitstring.

fun Con(bitstring,bitstring):bitstring.

reduc forall x:bitstring,y:bitstring;

Split(Con(x,y))=(x,y).

**Figure 2.** Definition and function declaration.

```
(*------------------------event------------------------------*)
event UAVLoginPhase(UAV).
event UAVAuthentication(UAV).
event UAVSessionKey(UAV).
event GSSSessionKey(GSS).
event GSSAuthentication(GSS).
(*------------------------queries------------------------------*)
query attacker(SKu).
query attacker(SKg).
query attacker(sk_gss).
query attacker(di).
query inj-event(UAVAuthentication(uav)) ==> inj-event(UAVLoginPhase(uav)).
query inj-event(GSSAuthentication(gss)) ==> inj-event(UAVAuthentication(uav)).
query inj-event(GSSSessionKey(gss)) ==> inj-event(GSSAuthentication(gss)).
query inj-event(UAVSessionKey(uav)) ==> inj-event(GSSSessionKey(gss)).
```

**Figure 3.** Events and queries.

```
(*---------------------UAV's process--------------------*)
let
UAVProcess(UIDi:bitstring,bioi:bioinformation,Ei:bitstring,Gi:bitstring,Hi:bitstring,taoi:bitstring,
PKgssj:bitstring)=
    let sigmai=Rep(bioi,taoi) in
    let Ai=XOR(Hash(sigmai),Ei) in
    let PIDi=Hash(Con(UIDi,Ai)) in
    let di=XOR(Gi,Hash(Con(Ai,PIDi))) in
    if Hi=Hash(Con(Ai,Con(di,PIDi))) then
        event UAVLoginPhase(uav);
    new rci:nonce;
    new Time1:timestamp;
    let ci=bit_nonce(rci) in
    let T1=bit_timestamp(Time1) in
    let Pit=EccMul(ci,PKgssj) in
    let Pi=EccMul(ci,P) in
    let W1=XOR(PIDi,Hash(Con(GIDj,Pit))) in
    let W2=XOR(di,Hash(Con(PIDi,Con(T1,Pit)))) in
    let W3=Hash(Con(PIDi,Con(di,Con(Pit,T1)))) in
    out(chn,(W1,W2,W3,Pi,T1));
    event UAVAuthentication(uav);
    in(chn,(W4:bitstring,W5:bitstring,W6:bitstring,Kj:bitstring,T2:bitstring));
    let nKit= EccMul(ci,Kj) in
    let TIDi=XOR(W4,Hash(Con(nKit,PIDi))) in
    let nQj=XOR(W5,Hash(nKit)) in
    let SKu=Hash(Con(nKit,Con(PIDi,TIDi))) in
    if W6=Hash(Con(SKu,Con(nQj,T2))) then
        event UAVSessionKey(uav).
```

**Figure 4.** UAV authentication process.

```
(*-----------------------GSS's process---------------------*)
let GSSProcess(GIDj:bitstring,chaj:bitstring,Yj:bitstring,Vj:bitstring,PKra:bitstring)=
    in(chn,(W1:bitstring,W2:bitstring,W3:bitstring,Pi:bitstring,T1:bitstring));
    let resj=PUF(chaj) in
    let sk_gss=XOR(Yj,Con(GIDj,resj)) in
    let nPit=EccMul(sk_gss,Pi) in
    let nPIDi=XOR(W1,Hash(Con(GIDj,nPit))) in
    let ndi=XOR(W2,Hash(Con(nPIDi,Con(T1,nPit)))) in
    let nAi=EccSub(EccMul(di,P),EccMul(nPIDi,PKra)) in
    if W3=Hash(Con(nPIDi,Con(ndi,Con(nAi,Con(nPit,T1))))) then
        event GSSAuthentication(gss);
    new rej:nonce;
    new rnj:nonce;
    new Time2:timestamp;
    let ej=bit_nonce(rej) in
    let nj=bit_nonce(rnj) in
    let T2=bit_timestamp(Time2) in
    let Kit=EccMul(ej,Pi) in
    let Kj=EccMul(ej,P) in
    let TIDi=XOR(nPIDi,Hash(nj)) in
    let W4=XOR(TIDi,Hash(Con(Kit,nPIDi))) in
    let SKg=Hash(Con(Kit,Con(nPIDi,TIDi))) in
    let k=XOR(Vj,Hash(Con(GIDj,sk_gss))) in
    let Qj=Hash(Con(TIDi,k)) in
    let W5=XOR(Qj,Hash(Kit)) in
    let W6=Hash(Con(SKg,Con(Qj,T2))) in
    event GSSSessionKey(gss);
    out(chn,(W4,W5,W6,Kj,T2)).
```

**Figure 5.** GSS authentication process.

```
---------------------------------------------------------
Verification summary:
Query not attacker(SKu[]) is true.
Query not attacker(SKg[]) is true.
Query not attacker(sk_gss[]) is true.
Query not attacker(di[]) is true.
Query inj-event(UAVAuthentication(uav[])) ==> inj-event(UAVLoginPhase(uav[])) is true.
Query inj-event(GSSAuthentication(gss[])) ==> inj-event(UAVAuthentication(uav[])) is true.
Query inj-event(GSSSessionKey(gss[])) ==> inj-event(GSSAuthentication(gss[])) is true.
Query inj-event(UAVSessionKey(uav[])) ==> inj-event(GSSSessionKey(gss[])) is true.
---------------------------------------------------------
```

**Figure 6.** Results.

## 6. Performance Comparison

This section presents a detailed performance evaluation of our proposed scheme, focusing on computation, communication, and security aspects. It is benchmarked against significant existing works, namely Kumar et al. [16], Son et al. [17], Kwon et al. [20], and the Babu et al. approach [19].

### 6.1. Computation Cost

To compare computational costs, Table 6 provides a detailed analysis, contrasting our proposed scheme with the aforementioned studies. This assessment was conducted on a personal computer equipped with an Intel(R) (Intel, Santa Clara, CA, USA) Core(TM) i5-1035G1 CPU @ 1.00 GHz (1.19 GHz), 16.0 GB RAM, and a Windows 10 64-bit operating system. The evaluation measures the computation times for cryptographic one-way hash

functions, elliptic curve point operations, and bilinear pairing functions, recorded as $T_h$, $T_m$, and $T_b$, respectively. These times are 0.056 milliseconds for $T_h$, 2.806 milliseconds for $T_m$, and 6.892 milliseconds for $T_b$.

Our scheme primarily focuses on handover authentication overhead, as the initial authentication occurs only once. In this context, the computation cost for a UAV in our scheme is $5T_h$, and for GSS nodes, it is $9T_h$. The total computation cost thus approximates to $5T_h + 9T_h \approx 0.784$ milliseconds. In contrast, the total computation costs for the approaches in Kumar et al. [16] and Son et al. [17] are approximately 40.654 milliseconds and 0.84 milliseconds, respectively. The schemes by Kwon et al. [20] and the Babu et al. approach [19] require about 12.344 milliseconds and 1.568 milliseconds, respectively. Figure 7 visually represents these findings. A comparative analysis highlights the lower computational overhead of our proposed scheme relative to its counterparts.

**Table 6.** Comparison of Computation Cost.

| Scheme | Device | Infrastructure | Total Performed Operation |
|---|---|---|---|
| [16] | $4T_m + 9T_h + 1T_b$ | $3T_m + 2T_h + 2T_b$ | $7T_m + 11T_h + 3T_b \approx 40.654$ ms |
| [17] | $6T_h$ | $9T_h$ | $15T_h \approx 0.84$ ms |
| [20] | $2T_m + 7T_h$ | $2T_m + 13T_h$ | $4T_m + 20T_h \approx 12.344$ ms |
| [19] | $17T_h$ | $11T_h$ | $28T_h \approx 1.568$ ms |
| Proposed | $5T_h$ | $9T_h$ | $14T_h \approx 0.784$ ms |

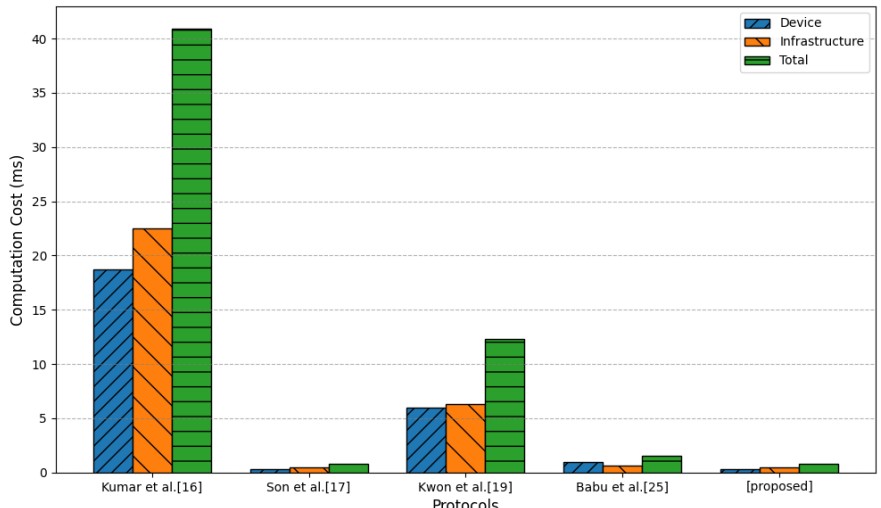

**Figure 7.** Comparison of computation cost.

*6.2. Communication Cost*

The communication costs of the proposed scheme were evaluated during the authentication phase, Table 7 provides a detailed analysis. We assumed the following sizes for various elements: timestamps, identities, and random numbers at 32 bits, 160 bits, and 160 bits respectively, encryption/decryption processes at 256 bits; and hash function outputs at 256 bits. The elliptic curve point size P = (Px, Py) was considered to be 320 bits. In the UAV authentication phase, a UAV and a GSS exchange two messages, with sizes detailed as follows: $Msg_1 = \{TID_i, M_7, M_8, M_9, T_3\} = (256 + 256 + 256 + 256 + 32) = 1056$ bits and $Msg_2 = \{M_{10}, M_{11}, M_{12}, T_4\} = (256 + 256 + 256 + 32) = 800$ bits. The total communication cost thus amounts to 1056 + 800 = 1856 bits.

For comparison, the communication costs in the schemes of Kumar et al. [16], Son et al. [17], Kwon et al. [20], and the Babu et al. approach [19] are 3200 bits, 2112 bits, 2560 bits, and 2784 bits, respectively. Figure 8 visually compares the proposed scheme's communication overhead with these other schemes, highlighting that the proposed scheme is competitive in terms of communication costs.

**Table 7.** Comparison of Communication Cost.

| Scheme | Number of Transmitting Messages | Communication Overhead (in Bits) |
|---|---|---|
| [16] | 4 | 3200 |
| [17] | 2 | 2112 |
| [20] | 4 | 2560 |
| [19] | 3 | 2784 |
| Proposed | 2 | 1856 |

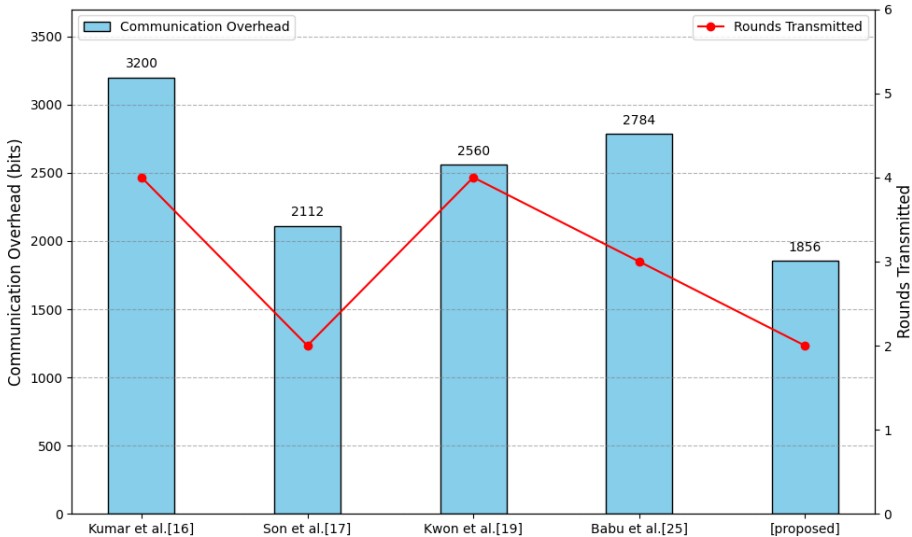

**Figure 8.** Comparison of communication cost.

### 6.3. Security Features

Table 8 offers a comprehensive analysis of the security functionalities comparing our newly developed protocol with earlier versions. This assessment underscores critical security aspects, such as "Mutual Authentication", "Impersonation Attack Resistance", "Replay Attack Defense", "Protection against Device Physical Capture", "Mitigation of Man-In-The-Middle (MITM) Attacks", "Ensuring Anonymity and Untraceability", and "Guaranteeing Perfect Forward Secrecy". As elaborated in Section 4, our protocol not only incorporates these security measures but also surpasses in their practical application. In contrast, the previously established protocols [16,17,20], and [19] either overlook these critical security dimensions or are inadequate in their assurance. Our protocol, by catering to a broader spectrum of potential threats in wireless channels, significantly bolsters security over the existing models.

**Table 8.** Comparison of Security Features.

| Security Features | [16] | [17] | [20] | [19] | Ours |
|---|---|---|---|---|---|
| Mutual Authentication | ✓ | ✓ | ✓ | ✓ | ✓ |
| Impersonation Attack | ✓ | × | × | ✓ | ✓ |
| Replay Attack | ✓ | ✓ | ✓ | ✓ | ✓ |
| Device Physical Capture Attack | × | × | × | × | ✓ |
| MITM Attack | ✓ | ✓ | ✓ | ✓ | ✓ |
| Anonymity and Untraceability | ✓ | ✓ | ✓ | ✓ | ✓ |
| Perfect Forward Secrecy | ✓ | ✓ | ✓ | ✓ | ✓ |

## 7. Conclusions

In this paper, we developed and introduced a lightweight secure handover authentication protocol tailored for UAV applications. This protocol incorporates an initial au-

thentication phase when a UAV enters a GSS domain and efficiently uses information from previous authentications to streamline the authentication process during transitions to subsequent GSS domains. This strategy significantly optimizes operational efficiency in dynamic environments for UAVs. The protocol demonstrates exceptional resistance to a variety of security threats, including UAV hijacking, identity spoofing, and replay attacks, thereby underscoring its reliability in securing UAV communications. The security and robustness of our protocol have been rigorously validated using the ROR model and the ProVerif tool. Moreover, our comprehensive performance analysis shows that our protocol surpasses existing solutions in significantly reducing computational and communication overheads, reflecting its specialized optimization for UAV scenarios.

The impact of our work extends beyond the direct benefits of improved authentication efficiency and security. By addressing the unique challenges of UAV handover scenarios, our protocol contributes to broader efforts to enhance the integrity of UAV operations in increasingly complex airspace environments. This research not only lays a solid foundation for safer and more efficient UAV deployments but also provides important insights for future studies on advanced authentication mechanisms, further refining UAV communication and operation protocols.

**Author Contributions:** Conceptualization, K.W. and S.W.; methodology, K.W. and Y.W.; software, K.W. and J.W.; validation, K.W., S.W. and L.H.; formal analysis, K.W. and Q.X.; investigation, K.W. and S.W.; resources, K.W., S.W. and J.W.; writing—original draft preparation, K.W. and S.W.; writing—review and editing, K.W. and S.W.; visualization, K.W. and L.H.; supervision, S.W. and Q.X. All authors have read and agreed to the published version of the manuscript.

**Funding:** This work was supported by the National Natural Science Foundation of China under Grant U21A20466 and the Hangzhou Joint Fund of the Zhejiang Provincial Natural Science Foundation of China under Grant No. LHZSZ24F020002.

**Data Availability Statement:** Data are contained within the article.

**Conflicts of Interest:** The authors declare no conflicts of interest.

## Abbreviations

The following abbreviations are used in this manuscript:

| | |
|---|---|
| $RID$ | Identity of $RA$ |
| $UAV_i$ | i-th UAV |
| $UID_i$ | Identity of $UAV_i$ |
| $PID_i$ | Pseudo-identity of $UAV_i$ |
| $GSS_i$ | i-th GSS |
| $GID_i$ | Identity of $GSS_i$ |
| $E_p(a,b)$ | An elliptic curve |
| $P$ | Generator of $\mathbb{G}$ |
| $TID_i$ | Temporary identity of $UAV_i$ |
| $PK_{RA}, SK_{SA}$ | Public key and private key of RA |
| $SK_{it}$ | Session key of $UAV_i$ and $GSS_i$ |
| $SK_{GSS_i}$ | Private keys of $GSS_i$ |
| $PK_{GSS_i}$ | Public keys of $GSS_i$ |
| $Gen(.)$ | The generating function of Fuzzy extractor |
| $Rep(.)$ | The reproduction function of Fuzzy extractor |
| $PUF()$ | Physical unclonable function |
| $bio_i$ | The biological information of user |
| $\sigma_i, \tau_i$ | Biological key and auxiliary parameter |
| $cha_i, res_i$ | The challenge and response of the PUF in $CS_i$ |
| $n_i, n_j$ | Random nonces |
| $T_*$ | Timestamp |
| $h(\cdot)$ | One-way hash function |
| $\oplus$ | Exclusive OR operation |

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
