# Peer review of "A Secure Authentication Protocol Supporting Efficient Handover for UAV"

_mathematics, doi:10.3390/math12050716_

Round 1

Reviewer 1 Report

Comments and Suggestions for Authors

1.      It is not clear from the abstract how anonymous measures and PUF were incorporated into the proposed scheme. I suggest that the author state in clear terms the method adopted to achieve this.

2.      UAV on line 14 should be written in full.

3.      The contributions 2 and 3 on lines 59-67 are the fallouts of the contributions 1. They are not contributions per say. The author should remove them or reframe to reflect the actual contributions intended.

It is not clear what makes the presented model a lightweight. I suggest the author explain the features that makes the presented handover scheme a lightweight.

Reviewer 2 Report

Comments and Suggestions for Authors

1. The key contributions and novelty of the proposed work have not been highlighted with respect to existing works in literature.

2. There is an abundance of literature available in literature on the topics addressed in this work. How does the submitted work compare with the following existing works?

[R1] H. Khalid et al., "HOOPOE: High Performance and Efficient Anonymous Handover Authentication Protocol for Flying Out of Zone UAVs," in IEEE Transactions on Vehicular Technology, vol. 72, no. 8, pp. 10906-10920, Aug. 2023, doi: 10.1109/TVT.2023.3262173.

[R2] X. Ren et al., "A Novel Access and Handover Authentication Scheme in UAV-Aided Satellite-Terrestrial Integration Networks Enabling 5G," in IEEE Transactions on Network and Service Management, vol. 20, no. 3, pp. 3880-3899, Sept. 2023, doi: 10.1109/TNSM.2023.3246732. 

[R3] Tejasvi Alladi, Vinay Chamola,  Naren, Neeraj Kumar, PARTH: A two-stage lightweight mutual authentication protocol for UAV surveillance networks, Computer Communications, Volume 160, 2020, Pages 81-90, ISSN 0140-3664, https://doi.org/10.1016/j.comcom.2020.05.025.

3. It is evident from above the recent relevant works in literature have not been included.

4. Useful insights are missing from the submitted work.

Reviewer 3 Report

Comments and Suggestions for Authors

1. The need for the presence of figures 2, 3, 4, 5 in the text is not well justified due to the lack of clear explanations of the code. Presenting them as an appendix might be more suitable.
2. „Figure 9. Comparison of communication cost“ is present in the text, but it is not mentioned at all.
3. An important note is related to the conclusion. At just 10 lines, it doesn't sum up the paper's point well enough. It needs to be strengthened.
4. Figure 9. “Comparison of communication cost“ is present in the text, but it is not mentioned at all. The same goes for Figure 8. All figures must be referenced.
5. There cannot be two figures with the same title „Results“ - Figure 6. and Figure 7.

Other remarks
1. Missing explanations of some abbreviations, like ROR. All abbreviations need to be checked for explanations. I also recommend not to use an abbreviation in the title
2. ProVerif must be spelled the same way everywhere.
3. In Table 9, instead of Security Features from A1 to A7, it is better to include the names of the attacks: „Mutual Authentication“, „Impersonation Attack“, etc. Also, the font size in the table is too large and should be reduced.

Round 2

Reviewer 2 Report

Comments and Suggestions for Authors

The authors have addressed all concerns of the reviewer.

Reviewer 3 Report

Comments and Suggestions for Authors

I am satisfied with the changes.